# Universal Preparation Strategy for Ultradurable Antibacterial Fabrics through Coating an Adhesive Nanosilver Glue

**DOI:** 10.3390/nano12142429

**Published:** 2022-07-15

**Authors:** Jundan Feng, Lingling Feng, Sijun Xu, Chunhong Zhu, Gangwei Pan, Lirong Yao

**Affiliations:** 1National & Local Joint Engineering Research Center of Technical Fiber Composites for Safety and Protection, Nantong University, Nantong 226019, China; feng5017@hotmail.com (J.F.); feng20220606@hotmail.com (L.F.); pangangwei@ntu.edu.cn (G.P.); ylr8231@ntu.edu.cn (L.Y.); 2Faculty of Textile Science and Technology, Shinshu University, Nagano 386-8567, Japan; zhu@shinshu-u.ac.jp

**Keywords:** nanosilver, fabric, antimicrobial, washable, reusable

## Abstract

Microbiological protection textile materials played an important role in the battle against the epidemic. However, the traditional active antimicrobial treatment of textiles suffers from narrow textile applicability, low chemical stability, and poor washability. Here, a high-strength adhesive nanosilver glue was synthesized by introducing nontoxic water-soluble polyurethane glue as a protectant. The as-prepared nanosilver glue could adhere firmly to the fiber surfaces by forming a flexible polymer film and could encapsulate nanosilver inside the glue. The as-prepared nanosilver had a torispherical structure with diameter of ~22 nm, zeta potential of −42.7 mV, and good dispersibility in water, and it could be stored for one year. Further studies indicated that the nanosilver glue had wide applicability to the main fabric species, such as cotton and polyester fabric, surgical mask, latex paint, and wood paint. The antimicrobial cotton and polyester fabrics were prepared by a simple impregnation–padding–baking process. The corresponding antimicrobial activity was positively correlated with nanosilver content. The treated fabrics (500 mg/kg) exhibited ultrahigh washing resistance (maintained over 99% antibacterial rates for 100 times of standard washing) and wear resistance (99% antibacterial rates for 8000 times of standard wearing), equivalent breathability to untreated fabric, improved mechanical properties, and good flexibility, demonstrating a potential in cleanable and reusable microbiological protection textiles.

## 1. Introduction

Public health and medical services have faced enormous challenges in recent years. Bioprotective fibers and related textile materials played a key role in the battle against the epidemic but also reflected their shortcomings [1]. Traditional disposable nonwoven textiles cannot block all pathogens or kill them because pathogens can multiply on their fiber surfaces. Used and discarded clothing may even become a vector of pathogen transmission [2]. Therefore, cleanable and reusable textiles with active antimicrobial properties are urgently needed [3]. The active antimicrobial properties of textiles are usually obtained with the introduction of antimicrobial agents, which can be divided into organic, inorganic, and natural agents [4,5,6]. The most used inorganic antimicrobial materials are metals, especially nanosilver, due to their superiority to organic agents, including ultrahigh and broad-spectrum antimicrobial activities, low bacterial resistance, and good biosafety [7,8,9]. 

However, the application of nanosilver in antimicrobial textiles should meet the following requirements, such as high water solubility to achieve functional finishing of textiles by traditional dyeing equipment, wide applicability to mainstream textile fibers, no obvious decrease in the functional treatment on the wearing comfortableness of the fabric, and good oxidation and washing resistance to meet the cleanable and reusable needs. Such properties depend on the physical and chemical properties of the nanosilver colloidal solution, which are mainly determined by their capping molecules [10]. Nanosilver allows for attachment of molecules on its surface through specific physical and chemical sorption, which should be attributed to their high surface chemical activity and high surface-to-volume ratio [11,12]. The functionalization of molecules is due to the following conditions. Core–shell structured nanosilver can be stabilized against agglomeration and chlorination/oxidation by coating the outer layers of organic molecules. The molecules on the nanosilver surface provide electrostatic, steric, and electrosteric repulsion for the nanosilver, imparting it with good solution stability. The good water stability of nanosilver is extremely important for the applications in biomedicine, water treatment, and functional textiles [13,14,15,16]. The surface charges of nanosilver are tunable by surface modification with cationic or anionic molecules [17,18]. The most important purpose for surface functionalization of nanosilver with molecules is imparting new functions. Molecule-capped nanosilver can be used with two opposite functional goals. The first goal is utilizing the nanoproperties of nanosilver, where the capping molecules only serve as protection agents to prevent nanosilver from agglomeration and chlorination/oxidation [19]. The second goal is that the nanosilver core and organic shell contribute to the functions of nanosilver [20,21]. In this case, nanosilver inherits the properties of nanosilver, the capping molecule, and even spawns new functions. 

In the case of nanosilver functionalized fabrics, the capping molecules should possess good protection ability of nanosilver against chemical reagents and high combining capacity of fabrics to achieve good durability without damaging the wearability of the fabric. In our previous studies, we designed hyperbranched polymer-modified silver nanoparticles that can spontaneously adsorb and adhere to the fiber surface through electrostatic adsorption and hydrogen bonding interactions [22]. However, this strategy still suffers from poor washing fastness and wear resistance, evidencing that only van der Waals forces and hydrogen bonding forces between small molecules and fiber-forming polymers cannot resist the invasion of detergents. Here, we further developed a high-strength adhesive nanosilver glue by introducing nontoxic and environmentally friendly water-soluble polyurethane glue as a capping agent. Polyurethane glue has serval advantages, including excellent protection ability of nanosilver due to its amphiphilic properties, high complexation ability of amido groups with silver, and strong adhesion to mainstream fibers, such as cotton, silk, and polyester fibers [23]. The as-prepared nanosilver glue can adhere firmly to the fiber surfaces by forming a flexible polymer interpenetrating network film and encapsulate nanosilver in polyurethane-interpenetrating network, isolating it from oxygen and chemicals, preventing it from physically exfoliating from the fiber surface, such as friction, and avoiding direct contact with the human body. Our studies showed that nanosilver glue is applicable to the main fabric species, such as cotton and polyester fabrics and surgical masks, and can be mixed with latex paint and wood paint for antimicrobial finishing of buildings and furniture. The as-prepared coated fabrics (500 mg/kg) exhibited ultrahigh washing resistance (100 times of standard washing) and wear resistance (8000 times), with improved mechanical and waterproof properties, indicating the potential application of nanosilver glue in microbial protection materials.

## 2. Materials and Methods

### 2.1. Materials

Cotton and polyester fabrics were acquired from Zhangjiagang Nellnano Technology Co., Ltd. (Suzhou, China). Silver nitrate (AgNO_3_) was purchased from Shanghai Institute of Fine Chemical Materials (Shanghai, China). Water-soluble polyurethane (anionic aliphatic polyether type, viscosity of more than 250 mPa·S at 25 °C, solid content of 38 ± 2%, PH of 6–7 at 25 °C) was obtained from Qiancheng Plastic Chemical Materials Co., Ltd. (Qingdao, China). Artificial sweat was provided by Dongguan ChangFeng Automation Technology Co., Ltd. (Dongguan, China). Sodium borohydride (NaBH_4_), nitric acid (HNO_3_), and ammonia were purchased from Sinopharm (Beijing, China). Gram-negative Escherichia coli (*E. coli*, ATCC 25922) and Gram-positive Staphylococcus aureus (*S. aureus*, CMCC26003) were obtained from Shanghai Luwei Technology Co., Ltd. (Shanghai, China). Nutrient agar and nutrient broth were purchased from Shanghai Zhongke Insect Biotechnology Development Co., Ltd. (Shanghai, China). Phosphorus-free ECE detergent was purchased from Xiangda Instrument Co., Ltd. (Guangzhou, China).

### 2.2. Synthesis of Nanosilver Glue

Ammonia was gradually titrated into 10 mL of 7.87 g/L AgNO_3_ solution until the mixture changed from earthy yellow to colorless to study the influence of polyurethane concentration on the size and zeta potential of silver nanoparticles in glue. Then, 0.1–3 g of 300 g/L aqueous polyurethane solution was mixed with the AgNO_3_ ammonia complex solution and stirred for 30 min. NaBH_4_ (1 g/L) was slowly added to the above mixture through stirring until the color of the solution stopped changing. The resultant nanosilver glue was further added with water to achieve 50 mL solution, stored in a brown colored bottle, and kept in a dark place. 

An aqueous polyurethane solution was mixed with 10–40 mL of 7.87 g/L AgNO_3_ ammonia complex solution and then reduced by NaBH_4_ as described above. The mass ratio of polyurethane and silver was set as the optimal value determined by the above study. The resultant nanosilver glue was further added with water to achieve the set volume (50 mL) solution, stored in a brown colored bottle, and kept in a dark place. 

### 2.3. Universal Preparation Process for Ultradurable Antimicrobial Fabrics

The as-prepared nanosilver glue was applicable to most fiber surfaces. In this study, cotton and polyester fabrics were chosen as the typical models of natural fiber and chemical fiber fabrics, respectively, and the typical coating process of nanosilver glue to ultradurable antimicrobial fabrics was described as follows: 

Cotton and polyester fabrics were washed with deionized water and ethanol several times to remove the impurities and dried at 60 °C for 1 h. The test fabric was then impregnated with nanosilver glue for 30 min with silver concentration ranging from 10 mg/L to 4000 mg/L and a bath ratio of 1:50. The test fabric was padded with a small laboratory padder (Y571GC, Laizhou Yuanmore Instrument Co., Ltd., Laizhou, China) and baked at 60 °C for 60 min and at 120 °C for 10 min to induce the cross-linking reaction.

### 2.4. Characterization

The morphological structure and distribution of the nanosilver glue were observed by using a transmission electron microscope (TEM; JEOL 2100F, Tokyo, Japan) at an acceleration voltage of 120 kV. The surface morphologies and chemical structure of samples were observed by using a field emission scanning electron microscope (SEM; Gemini SEM 300, Carl Zeiss, Jena, Germany) and a SEM (JSM-6510, JEOL, Tokyo, Japan) equipped with an energy dispersive spectroscopy detector (EDS; X-act, Oxford Instruments, Oxford, UK). UltraSTEM200X. The absorbance and concentration of the nanosilver glue were recorded by using an ultraviolet (UV) spectrophotometer (UV-vis; TU-1901, Beijing Puxi General Instrument Co., Ltd., Beijing, China). The zeta potential and size distribution of the nanosilver glue were measured by using a Zetasizer (90 plus Zeta, Brookhaven Instruments Corporation, Holtsville, NY, USA). The chemical structure of samples was analyzed through Fourier transform infrared (FTIR; Thermo Nicolet iS50, Waltham, MA, USA) spectroscopy at the 4000–400 cm^−1^ range and resolution of 2 cm^−1^ and X-ray photoelectron spectroscopy (XPS, Thermo Scientific K-Alpha+, Waltham, MA, USA). The X-ray diffraction (XRD) patterns of samples were obtained by using an X-ray diffractometer (ARL XTRA, Thermo Switzerland ARL Corp., Neuhofstrasse, Switzerland). The mechanical property test was conducted on an electronic fabric tensile tester (YG065, Laizhou Electronic Instrument Co., LTD, Laizhou, China), where the fabrics (rectangle shape, 350 mm × 50 mm) were stretched at a constant rate of 10 cm/min, and an intelligent fabric crease elasticity meter (YG(B)541E, Wenzhou Darong Textile instrument Co., Ltd., Wenzhou, China) in accordance with the standard GB/T 3819 vertical method. The breathability of fabrics was tested by using a digital fabric breathability tester (YG(B)416E, Yakai Instruments, Cangzhou, China). Bending rigidity was analyzed by using a fabric assurance by simple testing (FAST-2, CSIRO, Canberra, Australia). The silver concentration in aqueous solution was detected through inductively coupled plasma mass spectrometry (ICP-MS, PerkinElmer NexION 350, Waltham, MA, USA).

### 2.5. Antimicrobial Test

The antimicrobial activities of nanosilver glue were tested by using the Oxford cup inhibition zone method [24]. The sterilized nutrient agar was poured into two Petri dishes and allowed to solidify. The solution (1 mL) of *E. coli* (1–5 × 10^6^ cfu/mL) and *S. aureus* (1–5 × 10^6^ cfu/mL) was evenly placed on the two agar dishes. Three Oxford cups were placed on each culture dish, and 2 mL of deionized water, polyurethane solution, and nanosilver glue solution were added dropwise. The plates were placed in a biochemical incubator for 24 h at 37 °C. The diameter of the suppression ring was measured and recorded. The absorbance of the bacteria at 546 nm was tested with a UV–vis spectrophotometer to study the antimicrobial kinetics of nanosilver glue against *E. coli* and *S. aureus* after 0–24 h of culture. The effect of the silver concentration in glue on antimicrobial activity was studied by testing the absorbance of the bacteria at 546 nm after 6 h of incubation. 

The modified oscillation method (GB/T 20922.3-2008, China) was used to test the antimicrobial properties of cotton and polyester fabrics, as described in our previous paper [25]. The survival of bacteria adhered to fabric samples was also determined using the previously reported incubation method [25]. Gram-negative *E. coli* (ATCC 25922) and Gram-positive *S. aureus* (CMCC 26003) were selected as the test strains. 

### 2.6. Release Kinetics of Silver Ions

The time-dependent release of silver ions was measured as follows. An amount of 1 g nanoglue-coated fabric was impregnated with 50 mL artificial sweat, and then, 2 mL immersion liquid was collected at immersion times of 0.5, 1 h, 2 h, 4h, 6 h, 12 h, and 24 h. The concentration of silver ion in the impregnation liquid was measured using inductively coupled plasma mass spectrometry (ICP-MS).

### 2.7. Washability and Wear Resistance Test

The washing fastness of fabrics was measured in accordance with the GB/T 12490-2014 standard test method No.AS1 (China). The fabrics (10 cm × 10 cm) were washed with 150 mL of detergent (4 g of ECE detergent dissolved in 1000 mL deionized water) in a programable washing fastness tester (SW-12J, Wenzhou Darong Textile instrument Co., Ltd., Wenzhou, China) at 40 °C for 30 min, then rinsed with deionized water, and dried at 60 °C. Laundering durability was evaluated by measuring the difference in the antimicrobial efficiencies of the fabric samples after repeated 50 and 100 times of washing cycles.

The wear resistance test of antimicrobial fabrics was conducted by using a Martindale friction testing machine in accordance with the GB/T 21196.2 standard test method (China), and the antimicrobial test of fabric samples was evaluated after 8000 times of standard wearing. 

## 3. Results and Discussion

Water-soluble polyurethane macromolecules were chosen as a glue molecule due to their good water solubility, high bonding strength to biomass and chemical fibers, excellent size limitation, and protection ability of nanosilver owing to the strong binding ability of electron-donating amide groups on the main chain to electrophilic nanosilver and good molecular flexibility. They were used to prepare a nanosilver glue suitable for textiles. Nanosilver glue was prepared through the reaction of AgNO_3_ and sodium borohydride in water with polyurethane glue as a protective agent (Figure 1). Nanosilver glue has strong adhesion to various textile fibers, including hydrophilic and hydrophobic fibers, and can form an adhesive flexible polymer protection film on fiber surfaces because of the amphiphilic nature and strong intermolecular forces, thereby protecting nanosilver from oxidation and achieving excellent washability and wear resistance of textiles.

The amount of polyurethane has an important effect on the size and surface charge of silver nanoparticles. As shown in Figure 1d,e, the average particle size of nanosilver glue solution (1 mg/mL) decreased from 78.7 nm to around 35 nm and then slightly increased to 48.2 nm when the polyurethane concentration increased from 2.4 g/L to 12 g/L and then to 24 g/L. The increased size of nanosilver at high concentration was due to the increased viscosity of the solution system impeding the diffusion of the reducing agent. Therefore, the optimal mass ratio of polyurethane to nanosilver was set at 1:9. At this mass ratio, increased silver concentration from 1 g/L to 4 g/L showed a decrease in particle size from 36 nm to around 22.5 nm and a slight decrease in the zeta potential from −42.2 mV to around −41 mV and slight increase to 25.6 nm when the concentration reached 5 g/L. In summary, the optimum synthesis process was achieved with silver concentration of 2–4 g/L and mass ratio of polyurethane of 1:9. 

Figure 1a,b show the typical morphology structure of the nanosilver glue (4 g/L) solution. Nanosilver showed a spherical morphology with average particle sizes around 22 nm. The particle size distribution of nanosilver consisted of two parts. One was silver nanoparticles with smaller particle size (10–30 nm), accounting for 88.3% of the total volume, and the other was larger silver nanoparticles (60–200 nm), accounting for 11.7% of the total volume (Figure 1h). Such particle size differences were also found by TEM. As shown in Figure 1a, a large number of nanoparticles with small particle size and a few much larger nanoparticles were observed in the nanosilver glue solution, consistent with the DLS result. The typical morphology and structure of the nanosilver glue solution (4 g/L of mother solution, diluted to 10 mg/L) are shown in Figure 1e. The surface potential and chemical stability of nanosilver were mainly determined in terms of the anionic polyurethane. As shown in Figure 1e, the typical surface potential of nanosilver was around −42.7 mV, indicating its good stability. As shown in Figure 1d, nanosilver had a typical characteristic absorption peak at 421 nm in the UV–vis range (325–575 nm), suggesting its nanometallic state [26]. The intensity of the absorption peak at 421 nm was linearly and positively correlated with the silver concentration in the range of 10–20 mg/L (Figure 1f). This law can be used to calculate the silver concentration. The as-synthesized nanosilver glue can be stored for one year without changing color, demonstrating the stability of the solution (Figure 1j). 

The polyurethane on the nanosilver surface gave other advantages to nanosilver, such as good film formation and toughness, which improved the durability of nanosilver coating. As shown in Figure 2c,d, a black and flexible film of nanosilver glue with silver content of 0.1 g/g was prepared by drying the nanosilver glue solution, which could be tailored to the desired shape. Nanosilver glue can be applied to a wide range of materials, such as emulsion paint, waterborne wood paint, and commercial mask, thereby suggesting its wide applicability.

The SEM images of nanosilver glue film are illustrated in Figure 3a,b. The film showed a smooth and even structure. Most silver nanoparticles in the nanosilver glue film were encapsulated inside the membrane, and only a few in the defect area were exposed outside (Figure 3b). The chemical structure was investigated on the basis of the FTIR spectra of polyurethane and nanosilver glue in Figure 3c. Water-soluble polyurethane can be regarded as a kind of block copolymer containing soft segment and hard segment. The soft segment consisted of oligomer polyethers, whose FTIR characteristic peaks were C–H stretching vibration at 2931 and 2866 cm^−1^ and C–O–C stretching vibration at 1101 cm^−1^ (Figure 3c). The hard segment mainly consisted of polyisocyanates, which were relatively rigid and stretched into rods at room temperature. The characteristic peaks were the stretching vibration of N–H at 3370 cm^−1^, deformation vibration of N–H at 1541 cm^−1^, and C=O stretching vibration at 1643 cm^−1^ of the carbonate bond [27]. For nanosilver glue, no major change was observed in the FTIR characteristic absorption peak compared with polyurethane, except for one imperceptible change. This change was the relatively decreased signal of N–H bending vibration in the nanosilver glue at around 1540 cm^−1^ due to the chemical interactions between electron acceptor metallic Ag and electron-donating amide groups [28]. This finding indicates the good protection performance of polyurethane. The wide-scan XPS spectra suggested that polyurethane and nanosilver glue contained C1s, N1s, and O1s signals, which were located at approximately 284, 398, and 532 eV, respectively (Figure 3e). The difference was that the nanosilver glue exhibited additional Ag3d signals at around 368 eV. The difference between the fitted 3d_5/2_ and 3d_3/2_ peaks of the Ag3d signal (approximately 6.0 eV) was equal to the standard value of Ag^0^ (6.0 eV) (Figure 3e), further indicating the metallic state of nanosilver in films [29].

*E. coli* and *S. aureus* were chosen as typical model strains to evaluate the antimicrobial activities of nanosilver glue in this experiment. Deionized water and polyurethane were used as controls. Figure 4a,b show the images of the inhibition zones of deionized water, polyurethane, and nanosilver glue on the *E. coli*- and *S. aureus*-inoculated nutrient agar surfaces. No inhibition zone was found around the Oxford cups of deionized water and polyurethane, indicating the nonantimicrobial effects of polyurethane. By contrast, an *E. coli* and *S. aureus* inhibition zone with diameters of around 3 and 5 mm was measured around the Oxford cup of nanosilver glue. This finding proved the good antibacterial activity of nanosilver against *E. coli* and *S. aureus*. The antibacterial activity was proven to depend on the nanosilver concentration. As shown in Figure 4c, the absorbance of *E. coli* and *S. aureus* at 546 nm after 6 h of incubation decreased significantly from 1.008 to 0.013 and from 0.877 to 0.027 when the concentration of nanosilver increased from 0.5 mg/L to 6 mg/L. No significant increase was observed in the antibacterial activity above 4 mg/L, implying the minimum inhibitory concentration. Figure 4d shows the inhibition growth kinetics of nanosilver glue against *E. coli* and *S. aureus* within the measurement time of 24 h. The absorbance of pure *E. coli* and *S. aureus* suspensions (controls) increased significantly at 1–3 h contact time and reached a peak at the 20 h contact time. However, the bacterial concentration of *E. coli* and *S. aureus* in the nutrient solution containing nanosilver glue (4 mg/L) remained mostly unchanged and even showed a slight decrease, suggesting certain bactericidal activities of nanosilver at an ultralow concentration. 

Figure 5a shows typical photos of cotton, nanosilver-coated cotton, polyester, and nanosilver-coated polyester fabrics. After treatment, the fabrics changed from white to yellow and showed an even color distribution, suggesting the uniform distribution of nanosilver. Note that the color of nanosilver glue could be covered by co-treatment of fabrics with a mixture of nanosilver glue and dyes (Appendix A). The surface morphology of fabrics was investigated through FESEM (Figure 5). Figure 5e–h show the low-magnification (×60) SEM images of the surfaces of cotton and polyester fabrics. For coated cotton and polyester fabrics, no polymer film was found to block the micron gaps formed by the warp and weft yarns, thereby ensuring the air and moisture permeability (Figure 5f,h). The fibers in the yarn were bound together by a polymer film in large quantities, as shown in Figure 5i–l. This condition may improve the resilience of the fabric to some extent. The surfaces of the nanosilver-coated cotton and polyester fibers were tightly covered by a polymer film, and no nanoparticles were observed. The polyurethane coating on the polyester fabric was smoother and had better structural integrity mainly due to the homogeneous smooth surface of the polyester fibers (Figure 5m–p). 

Nanosilver was believed to hide in two places. One was encapsulated inside the polyurethane film, and the other was embedded in the interface of fibers and polymer film due to the early physical adsorption. As shown in Figure 5q,r, numerous nanosilver was found to adhere to the fiber surfaces at the defects of the polyurethane coating. In addition, the DES maps of silver element in fabrics showed that the nanosilver was uniformly distributed in the fiber, indicating good protection of water-soluble polyurethane (Appendix A). 

Figure 6a,b show the FTIR spectra of cotton and polyester fabric samples. The C=O peak and N–H peak at 1707 and 1557 cm^−1^ for the coated cotton fabric can be attributed to the carboxyl and the amide groups of polyurethane (Figure 6a). The FTIR absorption characteristic of CH_2_ groups at 2856 cm^−1^ for the coated cotton fabric was stronger than that at 2920 cm^−1^, whereas the cotton fabric showed the opposite law [30]. The differences were due to the superposition effect of FTIR adsorption of the alkyl chain groups in polyurethane and cellulose. Similarly, the FTIR absorption intensity of C–H at 2845 cm^−1^ greatly increased in the coated polyester fabric compared with the pure polyester fabric, evidencing the adhesion of nanosilver glue. 

The chemical composition transformation of nanosilver-glue-coated cotton and polyester fabrics was further assessed through XPS. The typical wide-scan spectra of pure cotton and polyester fabrics and coated cotton and polyester fabrics are shown in Figure 6c. The characteristic peaks of C1s and O1s at approximately 284 and 532 eV were detected for all samples. By contrast, coated cotton and polyester fabrics had additional N 1s and Ag3d signals, suggesting the adhesion of nanosilver glue. In the Ag3d XPS spectra (Figure 6b,c), split Ag3d signals can be fitted to Ag3d_3/2_ at 373.8 eV and Ag3d_5/2_ at 367.8 eV in coated cotton and Ag3d_3/2_ at 373.23 eV and Ag3d_5/2_ at 367.17 eV in polyester fabrics, where the difference was around 6.0 eV, in line with bulk metallic Ag [29]. The good chemical stability was due to most nanosilver being encapsulated in the polyurethane coating, protecting it from oxidizing substances, such as oxygen. N1s XPS signals can directly reflect the existence of polyurethane because cotton and polyester are N-free polymers. As shown in Figure 5e,h, the deconvolution analysis of N1s peak indicated that the fabrics contained N–C=O and C–N bonds, which were derived from the amido linkage of polyurethane [31]. The C1s high-resolution XPS spectra indicated that the signals for pure cotton fabric can be decomposed to three peaks with binding energy of 284.8 eV (C–C/C–H), 286.2 eV (C–O), and 287.7 eV (C–O–C) [32]. After coating with amido-containing polyurethane, the relative intensity of C–O/C–N and C=O/C–O–C signals increased significantly. Similarly, the slightly increased relative intensity of C–O/C–N and decreased O=C–O signals for coated polyester fabric indicated the attachment of nanosilver glue.

Bacteria tend to adhere to traditional textiles, make clothes smell, and pose a health risk. The advantages of antibacterial nanosilver glue depend on low antibacterial concentration, broad spectrum, good chemical stability, and long-lasting properties and good biocompatibility because nanosilver was encapsulated and protected in polyurethane coating [33]. The principle of sterilization mainly depends on silver ion being released from the coating, which can contact and react with the bacterial cell, resulting in the destruction of the inherent components of bacteria or functional disorders and the death of bacteria [34]. 

The release of the silver element was divided into two stages. In the first stage, nanosilver was firstly oxidized and then released silver ions into the amorphous region of polyurethane coating. In the second stage, silver ions gradually penetrated into the external environment by osmotic pressure (Figure 7a). The release amount of silver ions initially increased rapidly, then entered the plateau region, and decreased slowly after 12 h incubation (Figure 7b). The peak concentrations of released silver were 0.128 mg/L for the nanosilver-glue-coated cotton fabric and 0.141 mg/L for the coated polyester fabric.

Contact sterilization experiments were performed by immersing the fabrics into *E. coli* and *S. aureus* culture solutions to assess the antibacterial property. As shown in Figure 8a–f, nanosilver-glue-coated cotton and polyester fabrics exhibited excellent antibacterial activities. The CFU of all cotton and polyester fabric samples declined by more than six orders of magnitude for *E. coli* and *S. aureus*, with corresponding bactericidal rates of up to 99.99% and inhibition rates of 99.999% when the content of nanosilver ranged from 300 mg/kg to 6000 mg/kg (Figure 8e,f). To determine the survival of bacteria on textile surfaces, the bacteria adsorbed on fabric surfaces were retrieved, and the survival of bacteria was determined by the classical phenotype colony counting method through growth on AGAR medium. Figure 9 displays that CFUs in the culture dishes of the bacterial eluent were zero for both polyester and cotton fabrics, indicating no viable bacteria on the fiber surfaces. Because the antibacterial activities of nanoglue-coated fabrics mainly came from their slow release of silver ions, the bacteria that came into contact with the antimicrobial coating were killed first. 

The outstanding washing durability is another important advantage of nanosilver-glue-coated fabrics. As shown in Figure 10a–f, the inhibition rate remained at 98.18 % for *E. coli* and 99.66% for *S. aureus* for 300 mg/kg of cotton fabric and 99.39% for *E. coli* and 99.89% for *S. aureus* for 300 mg/kg of polyester fabric under 50 times of standard washing (Figure 10e,f). The coated fabrics still maintained good antibacterial activities with the inhibition rate up to 97.5% for *E. coli* and 99.62% for *S. aureus* for 300 mg/kg of cotton fabric and 98.93% for *E. coli* and 99.76% for *S. aureus* for 300 mg/kg of polyester fabric even after 100 times of standard washing (Figure 11a–f). In addition, the inhibition rate of coated cotton fabrics for 100 times of washing increased from 99.64% to 99.999% for *E. coli* and from 99.9% to 99.999% for *S. aureus*, and that of coated polyester fabric increased from 99.86% to 99.999% for *E. coli* and from 99.95% to 99.999% for *S. aureus* when the nanosilver content increased from 500 mg/kg to 6000 mg/kg, indicating that the antibacterial activities of coated fabrics were still silver content dependent. These findings indicated that nanosilver glue has good adhesion to the cotton fabric with nanosilver content over 300 mg/kg. The nanosilver-glue-coated polyester fabric showed better antibacterial properties than cotton with the same silver content after washing. This condition was probably due to polyurethane exhibiting stronger bonding strength to polyester than to cotton and smoother and more homogeneous surfaces of polyester fibers than cotton fibers. The coated cotton and polyester fabrics (500 mg/kg) maintained over 99% antimicrobial activities for *E. coli* and *S. aureus* after 0, 50, and 100 times of washing, demonstrating the strong adhesion and stability of nanosilver coating. The coated cotton and polyester fabrics (500 mg/kg) worn up to 8000 times showed 99.999% antimicrobial activities for *E. coli* and *S. aureus* equal to those of untreated fabrics, suggesting the good wear resistance of nanosilver coating. The strong bonding interaction between nanosilver-embedded polyurethane coating and fabrics was responsible for the good washing resistance and wear resistance. TG test results showed that the thermal stability of both treated fabrics was slightly improved compared with untreated samples (Appendix A).

The mechanical properties of nanosilver-glue-coated fabrics were evaluated. As shown in Figure 12a,b, no decline in the air permeability of 500 mg/kg coated cotton and polyester fabrics and slight decrease were observed when the silver content increased from 500 mg/kg to 6000 mg/kg for the two coated fabrics. This condition was due to the nanosilver coating having no influence on the void ratio of interyarns, as evidenced by FESEM (Figure 5). By contrast, the tensile strength, acute elastic response angle, and bending rigidity of the coated cotton fabric had a positive correlation with the nanosilver content, which increased from 251.94 N, 73.11°, 0.0485 cN·cm to 301.68 N, 82.98°, 0.1608 cN·cm in the warp direction, and 295.18 N, 62.64°, 0.0668 cN·cm increased to 348.18 N, 73.84°, 0.1765 cN·cm in the weft direction, respectively (Figure 12c,g,k). The coated polyester fabric showed similar laws for the tensile strength, acute elastic response angle, and bending rigidity (Figure 12d,h,l). The water contact angle of the two coated fabrics was found to increase significantly by increasing the silver content (Figure 12i,j). The water contact angle of the coated cotton fabric increased from 109° to 137° and that of the coated polyester fabric increased from 114° to 135° when the silver content increased from 0 mg/kg to 6000 mg/kg. The tearing strength tests of cotton and polyester fabrics reached the maximum with nanosilver contents of 1000 and 500 mg/kg, respectively (Figure 12e,f). The changes in the physical and mechanical properties of the fabrics were due to the fibers within the yarn being bonded together by polyurethane, as shown in Figure 5j,l. This condition effectively improved the tensile, antiwrinkle, antibending, and hydrophobic properties of the cotton and polyester fabrics. The bending rigidity of cotton showed a slight increase for the cotton and an obvious increase for the polyester fabric (Figure 12k,l). However, the coated polyester fabric still maintained good flexibility (Appendix A). 

## 4. Conclusions

In this study, we successfully fabricated a type of nanosilver glue with high adhesion to various materials, such as fabrics and surgical masks, good compatibility with latex paint and varnish that can be used for coating walls and wood, and excellent antimicrobial properties for *E. coli* and *S. aureus*. The optimal mass ratio of polyurethane to silver for synthesizing nanosilver glue was 1:9, and the nanosilver concentration (2–5 g/L) had no obvious influence on the particle size and zeta potential. The as-prepared nanosilver glue showed a UV–vis adsorption peak at 421 nm, a spherical morphology with average particle sizes around 22 nm, a negative surface charge of around −42.7 mV, and a minimum inhibitory concentration of 4 mg/L. The nanosilver glue could form a tough and flexible film with nanosilver content up to 10% by using a simple drying process. The nanosilver in the formed film remained metallic and was endowed with good stretchability, toughness, and adhesion, which could be tailored to various shapes and stretched to ~50% of its original length. FESEM, FTIR, XPS, and TG tests showed that the nanosilver glue successfully adhered to cotton and polyester fiber surfaces without blocking the space between the yarns. Antibacterial tests suggested that 500 mg/kg of nanosilver-glue-coated cotton and polyester fabrics showed 99.999% antibacterial activities against *E. coli* and *S. aureus* and maintained over 99% even after 100 times of standard washing. Further studies exhibited that the air permeability remained unchanged for 500 mg/kg of cotton and polyester fabrics, and the tensile strength, tear strength, CRA, and hydrophobicity were improved by increasing the silver content. The bending rigidity of cotton showed a slight increase for cotton and an obvious increase for polyester fabric. However, the coated polyester fabric still maintained good flexibility. Our nanosilver glue showed high adhesion, wide applicability to various materials, and high washing resistance and wear resistance, showing its high potential in microorganism protection textiles.

## Data Availability

The data that support the findings of this study are available from the corresponding author upon reasonable request.

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
