# Peer review of "Universal Preparation Strategy for Ultradurable Antibacterial Fabrics through Coating an Adhesive Nanosilver Glue"

_nanomaterials, 2022, doi:10.3390/nano12142429_

Round 1
Reviewer 1 Report
This manuscript deals with the possibility to obtain an antibacterial nanosilver glue to protect for long time textiles from infection. The physical-chemical properties and the mechanical performances of the materials are provided as well as their ability to prevent infection (by both E. coli and S. aureus) has been proven even after a long aging period (i.e. washing steps).
The topic is of interest as textile infections can represent a dangerous spread of bacterial contamination thus for example extending hospitalization and exposing patients to further side effects in the clinical field.
The manuscript is well written as well as the experiments are logically designed and results are convincing. However, prior to suggest this work for publication some major concerns have to be considered:
1. The results of the direct infection of the treated textiles are convincing but I strongly suggest to add some images (for example by SEM) of the materials showing the lack of viable bacteria as sometimes results with textiles are overestimated due to the difficulty of detach or stain bacteria within fibers.
2. The major issue in this work is that the glue incorporating silver has been thought for the treatment of textile potentially in contact with skin or mucosa (for example the face mask) but a proper evaluation of the toxicity is lacking. Please note that silver amount is remarkable therefore a cytotoxicty test is required to prove the safety of the materials after the glue doping besides the antibacterial affect. Moreover, an irritation test is recommended due to the prolonged contact with human tissues.
Author Response
Reviewer 1:
This manuscript deals with the possibility to obtain an antibacterial nanosilver glue to protect for long time textiles from infection. The physical-chemical properties and the mechanical performances of the materials are provided as well as their ability to prevent infection (by both E. coli and S. aureus) has been proven even after a long aging period (i.e. washing steps).
The topic is of interest as textile infections can represent a dangerous spread of bacterial contamination thus for example extending hospitalization and exposing patients to further side effects in the clinical field.
The manuscript is well written as well as the experiments are logically designed and results are convincing. However, prior to suggest this work for publication some major concerns have to be considered:
1. The results of the direct infection of the treated textiles are convincing but I strongly suggest to add some images (for example by SEM) of the materials showing the lack of viable bacteria as sometimes results with textiles are overestimated due to the difficulty of detach or stain bacteria within fibers.
Response: Thank you very much for your suggestion! To verify whether the bacteria on the fabric surface are alive or dead, the bacteria adsorbed on the fabric surface are collected by the ultrasonic method after the fabric and bacteria are co-cultured for a certain period of time. Finally, the bacteria are cultured to calculate the number of bacteria and verify the number of bacteria on the fabric surface (Gao L, et al. Small. 2021;17(20):2006815). Our results showed that there were no living bacteria on the cotton and polyester fabrics, indicating the previous tests were reliable. The supplementary data are shown in Fig. 9 and Line 390-397, page 12.
2. The major issue in this work is that the glue incorporating silver has been thought for the treatment of textile potentially in contact with skin or mucosa (for example the face mask) but a proper evaluation of the toxicity is lacking. Please note that silver amount is remarkable therefore a cytotoxicty test is required to prove the safety of the materials after the glue doping besides the antibacterial affect. Moreover, an irritation test is recommended due to the prolonged contact with human tissues.
Response: Thank you very much for your suggestion! In fact, the as-prepared silver nanoparticle-capped fabric were used for bioprotective textiles as shown in the introduction, which was not in direct contact with skin or any human tissues. The as-prepared face mask shown in Figure 2 was consisted of three layers and nanosilver glue-coated viscose nonwoven fabric is located in the interlayer. In addition, the good biocompatibility of waterborne polyurethane-silver nanocomposites has already been proved by Hsu’ group and we have cited their paper ref. 33.
Reviewer 2 Report
The paper from Feng et al. reports on the preparation of a durable waterborne polyurethane coating containing silver torispherical nanoparticles, suitable for providing antibacterial properties to cotton and polyester fabrics. The manuscript is quite well written and the conclusions are quite well supported by the experimental data.
Minor remarks:
- please specify the coating dry add-on on the fabrics
- SEM analyses: it could be very useful to map Silver through EDS analysis, thus having a good idea of the distribution of silver nanoparticles
- finally, do the authors have an indication of the Tg (i.e., glass transition temperature) of the polyurethane glue?
Author Response
Reviewer 2:
The paper from Feng et al. reports on the preparation of a durable waterborne polyurethane coating containing silver torispherical nanoparticles, suitable for providing antibacterial properties to cotton and polyester fabrics. The manuscript is quite well written and the conclusions are quite well supported by the experimental data.
Minor remarks:
1. Please specify the coating dry add-on on the fabrics.
Response: Thank you very much for your suggestion! We have marked the coating add-on on the fabrics (Line128, page 3).
2. SEM analyses: it could be very useful to map Silver through EDS analysis, thus having a good idea of the distribution of silver nanoparticles.
Response: Thank you very much for your suggestion! We have added the EDS mapping and related analysis (Fig. S2 and Line 326-329, page 9).
3. Finally, do the authors have an indication of the Tg (i.e., glass transition temperature) of the polyurethane glue?
Response: Thank you very much! TG data was shown in Figure S1 (see Supplementary Information). In fact, TG cannot get the glass transition temperature and other data, but can only get corresponding epitaxial starting and termination temperatures, which have been marked in the Figure S3.
Since the thermal properties of the fabric change little before and after coating, the thermal properties was not analyzed in detail.
Reviewer 3 Report
The manuscript by Feng and coworkers, entitled "Universal preparation strategy for ultradurable antibacterial fabric through surface construction of an adhesive flexible polymer protection network by nanosilver glue" describes the development and characterization of composite coating with antibacterial properties and high adhesion to several materials. The manuscript is well structured. However, I still have some recommendations, which are presented below, prior publication in Nanomaterials:
- please mind the title of the manuscript in order to be more pointed towards the coating properties and/or application.
- why does the coated materials changes colour to yellow?
- will this colour change be accepted by the consumer?
- Page 9, line 302: How could the authors confirm the sentence: "Nanosilver was believed to hide in two places. One was encapsulated inside the polyurethane film, and the other was embedded in the interface of fibers and polymer film due to the early physical adsorption."
- Page 12, line 378: "TG test results showed that the thermal properties of both treated fabrics had no obvious change from those before treatment" why no changes could be observed? what do this result means?
Author Response
Reviewer 3:
The manuscript by Feng and coworkers, entitled "Universal preparation strategy for ultradurable antibacterial fabric through surface construction of an adhesive flexible polymer protection network by nanosilver glue" describes the development and characterization of composite coating with antibacterial properties and high adhesion to several materials. The manuscript is well structured. However, I still have some recommendations, which are presented below, prior publication in Nanomaterials:
1. Please mind the title of the manuscript in order to be more pointed towards the coating properties and/or application.
Response: Thank you very much for your suggestion! We have modified the title.
2. why does the coated materials changes color to yellow?
Response: Thank you! The yellow color of cotton and polyester fabrics was derived from the color of the nano-silver.
3. Will this colour change be accepted by the consumer?
Response: Thank you! The nano-silver glue can be dyed on the fabric together with the dye and the color of the nano-silver can be covered by the dye (See Figure S1, line 312-313).
4. Page 9, line 302: How could the authors confirm the sentence: "Nanosilver was believed to hide in two places. One was encapsulated inside the polyurethane film, and the other was embedded in the interface of fibers and polymer film due to the early physical adsorption."
Response: Thank you! We have added EDS analyze and FESEM data to prove our guess. As shown in Figures 5q and 5r, numerous nano-silver was found to attach to the fiber surface at the defects of the polymer coating. In addition, DES maps of silver in fabrics show that the silver element is uniformly distributed in the fiber, which proves that the nano-silver is uniformly distributed in the coating (Figures S2, line 326-330).
5. Page 12, line 378: "TG test results showed that the thermal properties of both treated fabrics had no obvious change from those before treatment" why no changes could be observed? What do this result means?
We came to this conclusion was because TG curve changed little before and after treatment and the corresponding epitaxial starting and termination temperatures changed a little. Nevertheless, because of the good thermal stability of polyurethane, the thermal stability has been slightly improved. Therefore, we revised the correlation analysis (line 433, 434).
Reviewer 4 Report
The work contains interesting results on the development of an effective method for creating textile materials with microbiological protection based on the use of silver nanoparticles. For this, a high-strength nanosilver adhesive was synthesized based on non-toxic water-soluble polyurethane adhesive as a protective agent. It has been shown that the glue can be successfully used for processing the main types of fabrics, as well as an additive for latex paint and wood paint. The resulting fabrics showed high resistance to washing and wear resistance, as well as a number of other useful consumer characteristics.
The work was carried out at a high scientific level using various physical and chemical means of analysis. It can be recommended for publication subject to the following remarks.
1. The antibacterial activity of silver nanoparticles, as is commonly believed, is mainly due to the toxic effect of silver ions released as a result of partial dissolution of particles upon contact with water, its vapor and during oxidation of the metal to silver oxide by oxygen. Therefore, it is desirable to provide data on the solubility of nanoparticles.
2. There are no data to compare the effect of nanosilver glue and glue with the inclusion of only silver ions.
3. In fig. 1f shows the results for Particle size distribution of nanosilver glue. However, they are not discussed in the text of the article. The presence of large particles with a size of approximately 100 nm is also indicated. Obviously, they were obtained by the DLS method. It is advisable to compare with the data obtained by the TEM method.
4. 99.999% antimicrobial activities for E. coli and S. aureus are reported. How did the authors achieve such values? Is it necessary to specify such accuracy?
Author Response
Reviewer 4:
The work contains interesting results on the development of an effective method for creating textile materials with microbiological protection based on the use of silver nanoparticles. For this, a high-strength nanosilver adhesive was synthesized based on non-toxic water-soluble polyurethane adhesive as a protective agent. It has been shown that the glue can be successfully used for processing the main types of fabrics, as well as an additive for latex paint and wood paint. The resulting fabrics showed high resistance to washing and wear resistance, as well as a number of other useful consumer characteristics.
The work was carried out at a high scientific level using various physical and chemical means of analysis. It can be recommended for publication subject to the following remarks.
1. The antibacterial activity of silver nanoparticles, as is commonly believed, is mainly due to the toxic effect of silver ions released as a result of partial dissolution of particles upon contact with water, its vapor and during oxidation of the metal to silver oxide by oxygen. Therefore, it is desirable to provide data on the solubility of nanoparticles.
Response: We have added time-dependent silver released data and related discussion was also given (Figure 7, line 373-383, page 11-12).
2. There are no data to compare the effect of nanosilver glue and glue with the inclusion of only silver ions.
Response: Thank you very much for your suggestion! Because silver ions have a positive charge, it is not compatible with water-soluble polyurethane. In addition, silver ions are usually not used as an antibacterial agent because they are dissolve in water, chemically unstable, and have a strong irritation to the skin.
3. In fig. 1f shows the results for Particle size distribution of nanosilver glue. However, they are not discussed in the text of the article. The presence of large particles with a size of approximately 100 nm is also indicated. Obviously, they were obtained by the DLS method. It is advisable to compare with the data obtained by the TEM method.
Response: Thank you very much for your suggestion! We added related discussion (line 215-221).
4. 99.999% antimicrobial activities for E. coli and S. aureus are reported. How did the authors achieve such values? Is it necessary to specify such accuracy?
Response: Thank you very much for your comments! The percentage reduction (cfu, %) was determined as follows:
cfu(%) =100*(C-A)/C (1)
where C and A are the bacterial colonies of the blank sample (without fabric) and antibacterial fabric, respectively.
It was necessary to specify such accuracy. Because in high grade biological laboratory such as P3 and P4 laboratory, antibacterial activities of the protective clothing requires more than 99.999%.
Reviewer 5 Report
The manuscript is dedicated to modification of the fabric species of different nature with polymer-stabilized antimicrobial silver nanoparticles. The research is well organised an the results are supported with the reliable data.
However one important correction should be done prior the publication.
Polyurethanes are insoluble in water. So the sample that was used in the research is not conventional polyurethane. According to the scheme 1 I suggest that it is block-copolymer with amphiphilic nature.
So, the data about this polymer: composition, molecular weight, ratio of different monomer units should be presented.
As a result, the section with FTIR analysis should be broadened with the data about groups of copolymer.
As well the amphiphilic nature of the polymer allows one to modify polyether and cotton fibers, so this fact should be mentioned in results/discussion.
Author Response
Reviewer 5:
The manuscript is dedicated to modification of the fabric species of different nature with polymer-stabilized antimicrobial silver nanoparticles. The research is well organised an the results are supported with the reliable data.
However one important correction should be done prior the publication.
1. Polyurethanes are insoluble in water. So the sample that was used in the research is not conventional polyurethane. According to the scheme 1 I suggest that it is block-copolymer with amphiphilic nature.
So, the data about this polymer: composition, molecular weight, ratio of different monomer units should be presented.
Response: Thank you very much for your suggestion! We have asked the company to provide about parameters. They said that the composition of polymer was anionic aliphatic polyether type, viscosity of over 250 mPa·S at 25℃, solid content of 38±2%, PH of 6-7 at 25℃ (line 95-96). They also said that waterborne polyurethanes usually measure the viscosity rather than molecular weight.
2. As a result, the section with FTIR analysis should be broadened with the data about groups of copolymer.
Response: Thank you very much for your suggestion! We have added related discussion about FTIR adsorption peaks of polyurethane (Figure 3c, Line 261-268, page 7).
3. As well the amphiphilic nature of the polymer allows one to modify polyether and cotton fibers, so this fact should be mentioned in results/discussion.
Response: Thank you very much for your suggestion! We have added related analyze (Line 196-200, page 5).
Round 2
Reviewer 1 Report
The Authors improved their manuscript according to the suggestions, therefore the manuscript ca be in my opinion accepted for publication.